# Implementation Rates and Predictors of Compliance with Enhanced Recovery After Surgery Protocols in Gynecologic Oncology: A Prospective Multi-Institutional Cohort Study

**DOI:** 10.3390/cancers17243991

**Published:** 2025-12-15

**Authors:** Vasilios Pergialiotis, Dimitrios Haidopoulos, Alexandros Daponte, Dimitrios Tsolakidis, Stamatios Petousis, Ioannis Kalogiannidis, Dimitrios Efthymios Vlachos, Vasilios Lygizos, Maria Fanaki, George Delinasios, Panagiotis Tzitzis, Philipos Ntailianas, Vasilios Theodoulidis, Georgia Margioula Siarkou, Nikoletta Daponte, Nikolaos Thomakos

**Affiliations:** 1First Department of Obstetrics and Gynecology, Alexandra Hospital, National and Kapodistrian University of Athens, 11528 Athens, Greece; dimitrioshaidopoulos@gmail.com (D.H.); vlachos.dg@gmail.com (D.E.V.); vlygizos@gmail.com (V.L.); maria.fanaki@gmail.com (M.F.); philiposntailianas@gmail.com (P.N.);; 2School of Health Sciences, Faculty of Medicine, University of Thessaly, 41500 Larissa, Greece; daponte@med.uth.gr (A.D.); gdelis94@gmail.com (G.D.);; 31st Department of Obstetrics & Gynecology, Aristotle University of Thessaloniki, “Papageorgiou” Hospital, 56429 Thessaloniki, Greecetheodoulidisvasilis@yahoo.gr (V.T.); 42nd Department of Obstetrics and Gynaecology, Aristotle University of Thessaloniki, 54622 Thessaloniki, Greece; 53rd Department of Obstetrics and Gynaecology, Aristotle University of Thessaloniki, 54124 Thessaloniki, Greece; 6Gynecologic Oncology Unit, Second Department of Obstetrics and Gynecology, Aristotle University, 54124 Thessaloniki, Greece

**Keywords:** enhanced recovery, ERAS, gynecologic oncology, adherence, predictors

## Abstract

This prospective multicenter cohort study evaluated adherence to Enhanced Recovery After Surgery (ERAS) protocols in gynecologic oncology during early implementation across five tertiary institutions. A total of 300 consecutive patients undergoing gynecologic cancer surgery were included. Compliance with predefined preoperative, intraoperative, and postoperative ERAS components was recorded using standardized forms. Optimal adherence was defined as fulfillment of at least 70% of applicable ERAS elements per patient. Overall, 70.3% of patients achieved optimal adherence; however, substantial interinstitutional variability was observed, with adherence rates ranging from 26.9% to 84.4%. High-volume centers demonstrated consistently higher compliance compared with lower-volume institutions. Preoperative components showed high uptake, whereas several intraoperative and early postoperative elements—particularly opioid minimization, tranexamic acid use, and early urinary catheter removal—had the lowest adherence. Multivariable analysis identified higher surgical complexity and poorer performance status as independent predictors of reduced adherence. These findings indicate that ERAS implementation is feasible but uneven, highlighting the need for structured workflows and multidisciplinary engagement, especially in complex procedures and low-volume centers.

## 1. Introduction

Enhanced recovery after surgery (ERAS) protocols transformed the perioperative management of patients in every aspect of modern surgery. The concept was first introduced by Henrik Kehlet in 1995 under the term fast-track surgery; however, its popularity was significantly delayed and the first evidence-based consensus protocol for ERAS was published by the ERAS Society^®^ in 2005, in reference to patients undergoing large bowel surgery [1]. In gynecologic oncology, the concept of ERAS was first introduced in 2016 [2] and society recommendations were provided in an updated form that was published in 2019 [3].

The revolutionary concept of ERAS is based on the implementation of a series of perioperative measures that help attenuate the physiological stress response to surgery, leading to improved postoperative homeostasis that results in accelerated functional recovery [4]. Specifically, ERAS protocols include a summary of recommendations that target the optimization of cardiovascular and pulmonary functions during the early postoperative period by emphasizing goal-directed fluid therapy, multimodal analgesia with opioid-sparing strategies, early mobilization, maintenance of normothermia, and early oral intake to support hemodynamic stability and preserve respiratory function [3].

Enhanced recovery after surgery programs have been adopted by multiple surgical specialties; however, adaptations in the core principles of perioperative optimization have been implemented among those to cover specialty driven needs. Therefore, despite the fact that the concept of ERAS is essentially the same among specialties, the variables in the different perioperative phases may differ substantially. Specifically, in colorectal surgery, ERAS guidelines highlight the need to avoid mechanical bowel preparation, early feeding, and ileus prevention strategies [1]. Perioperative protocols in hepatobiliary and pancreatic surgery emphasize the need for intraoperative fluid restriction, adequate glycemic control, and strict thermoregulation, which is essential in upper abdominal procedures that result in high metabolic stress [5]. In orthopedic surgery, fast functional recovery and immediate mobilization are essential and ERAS guidelines focus on parameters that enhance these functions, namely regional anesthesia and multimodal analgesia [6]. Likewise, thoracic ERAS pathways incorporate respiratory physiotherapy and structured chest-drain management to reduce pulmonary complications [7]. Despite the specialty-specific adaptations, all ERAS frameworks share the same goal of minimizing surgical stress, preserving physiologic stability, and expediting safe recovery.

The adoption of ERAS on an international level seems to differ and depend on the actual incorporation of a dedicated program [8]. The actual percentage of adoption of individual elements of the ERAS pathway ranges from 40% to 95%, with preemptive analgesia, antiemetic use, patient warming, and plan for opioid minimization being the parameters that are likely less adopted according to a recent meta-analysis that included 9076 participants [9]. In gynecology and gynecologic oncology, surgical drains also remain a critical component that seem to be disregarded as physicians seem to continue to be reluctant to adhering to current recommendations [10].

The effectiveness of ERAS protocols is directly related to the actual proportion of elements that were actually followed during the perioperative period. Previous research has shown that compliance rates that exceed the threshold of 70–80% of the sum of recommendations seem to result in a clinically meaningful benefit [11,12,13]. This is not, however, easy to achieve or even maintain, as significant coordination among members of the multidisciplinary team, namely surgeons, anesthesiologists, nursing staff, and other faculty is needed and must be followed up on a regular basis to ensure continuity in clinical practice [14,15].

Several factors may prevent physicians from successfully incorporating ERAS principles in individual patients. The most critical barriers of adherence are patient characteristics such as frailty, poor performance status, significant comorbidities, or complex surgical pathology that limits the feasibility of early mobilization or feeding [16,17]. Procedure-specific factors may also hinder adherence to ERAS protocols and result in suboptimal compliance. Among those, highly complex surgical procedures and prolonged operating times seem to be the predominant factors, with almost half of medium/high-complexity operations (47%) showing <75% ERAS compliance [18]. Nevertheless, the feasibility of ERAS has already been proposed, even among patients undergoing cytoreductive surgery [19].

In the present prospective multi-institutional study, we investigate compliance rates and predictors of compliance with ERAS components in a cohort of patients offered surgical management for gynecological cancer.

## 2. Methods

### 2.1. Study Design and Protocol Registration

The present prospective observational study is based on an interim analysis of the Enhanced Recovery Protocols in Gynecologic Oncology (ERGO) and aims to investigate the feasibility of and compliance to a structured ERAS protocol among Gynecological Oncological Centers in Greece. The study also aims to compare the outcomes among patients who fulfilled the minimum number of necessary criteria compared to those who were enrolled in ERAS protocols but did not meet the sufficient necessary criteria. The study is designed to run under real-world conditions, as a structured ERAS protocol had not been introduced in most participating institutions beforehand; hence, randomization was not feasible. The overall target enrollment is 600 patients, to be accrued over the period from October 2024 to July 2026. The present interim analysis is based on a sample of 300 participants.

All patients provided informed consent prior to participation in the present study, which was conducted in accordance with the Declaration of Helsinki, which ensures the ethical principles for medical research involving human participants and is designed considering the STROBE (Strengthening the Reporting of Observational Studies in Epidemiology) guidelines. The protocol was approved by the Institutional Review Boards of all participating centers and registered in the ClinicalTrials.gov database (identifier: NCT06655506).

### 2.2. Data Collection and Measured Variables

Data were collected prospectively using uniform electronic forms under the REDCap software managed by trained research coordinators at each center. Baseline patient characteristics were considered, including age, body mass index (BMI), smoking status, comorbidities (e.g., hypertension, diabetes), cancer site and stage (latest FIGO staging system), performance status using the ASA (American Society of Anesthesiologists) score and ECOG (Eastern Cooperative Oncology Group) score, and preoperative laboratory values. Intraoperative characteristics were also documented and included the surgical approach (laparoscopy vs. laparotomy), operative time, estimated blood loss, intraoperative transfusion requirement, complexity score of the procedure (using the Aletti surgical complexity score), and administered fluid volumes.

### 2.3. ERAS Protocol Implementation

The ERAS Society Guidelines for Gynecologic Oncology Surgery (2021 update) were considered as the reference framework. The selection of measured variables was based on predefined methodological criteria to ensure consistency with established ERAS implementation research. The ERAS pathway in gynecologic oncology comprised standardized elements from the preoperative, intraoperative, and postoperative care of gynecological cancer patients (Table 1). Briefly, preoperative items emphasize patient preparation, optimized fasting, carbohydrate loading, and evidence-based prophylaxis. Intraoperative measures focus on multimodal analgesia, normothermia, goal-directed fluid therapy, and minimizing tubes and drains. Postoperative components promote early oral intake, mobilization, catheter removal, multimodal analgesia, dietary advancement, glycemic control, and fluid optimization. Moreover, in our dataset we incorporated predictors commonly examined in prior multicenter ERAS audits, including age, BMI, comorbidities, performance status, surgical complexity, operative duration, and fluid management.

Each participating center included at least a dedicated anesthesiologist on the ERAS implementation team. Prior to study initiation, structured educational sessions were delivered to all anesthesiology departments, focusing on ERAS-aligned intraoperative practices, opioid-sparing analgesia, normothermia, and fluid therapy. These activities ensured protocol familiarity across specialties, allowing the study to evaluate real-world execution rather than gaps in training or agreement.

For exploratory analyses, participating centers were additionally grouped post-hoc into high- and low-volume categories based on their recruitment within the interim dataset. Centers contributing more than 60 patients were considered high-volume, whereas those enrolling fewer cases were classified as low-volume. This categorization was not predefined in the protocol and was used solely to contextualize inter-institutional variability in ERAS adherence.

### 2.4. Outcomes

Adherence to each ERAS item was documented as achieved or not achieved. The overall compliance rate was calculated as the percentage of elements fulfilled out of the total applicable ERAS components per patient. An arbitrary cut-off of 70% of fulfilled parameters was used to denote adequate adherence to the ERAS components per patient. Institutional compliance was determined by the mean adherence rate among patients treated in each center. This threshold is consistent with previous ERAS studies showing that adherence levels of approximately 70–80% are associated with clinically meaningful improvements in postoperative recovery and complication rates [11,18]. The primary outcome investigated in the present study referred to the actual compliance rates of ERAS principles, using the aforementioned cut-off value. Compliance rates per individual ERAS parameter were also considered as well as interinstitutional variability in ERAS implementation. The effect of time on the adherence of protocols was investigated after subgrouping cases per institution and evaluating differences in compliance rates per 30 cases included. The impact of patient characteristics (age, BMI, etc.) and intraoperative characteristics (Aletti score, surgical duration, etc.) on ERAS compliance was also evaluated.

### 2.5. Sample Size Calculation—Decision Point of Interim Analysis

Previous multicenter ERAS audits in gynecologic oncology have reported compliance rates ranging between 56% and 89% depending on institutional experience and resource availability [20,21,22]. Based on this evidence, a minimum sample of 300 patients was estimated to provide an 80% statistical power (1 − β = 0.8) to detect a 15% difference in overall compliance among participating centers, assuming a two-sided α = 0.05.

### 2.6. Statistical Analysis

Statistical analysis was performed using the SPSS 27.0 program (IBM Corp., Armonk, NY, USA). Evaluation of data distribution was performed with the Kolmogorov–Smirnov test and graphical methods. Continuous variables were presented as medians (interquartile range) and compared using the Mann–Whitney U or Kruskal–Wallis test, depending on the number of groups. Categorical variables were analyzed using the chi-square test or Fisher’s exact test, as appropriate.

A multivariate logistic regression analysis was conducted to identify factors associated with high ERAS compliance (≥70%) and favorable postoperative outcomes. The Backward stepwise method was applied, and odds ratios (OR) with 95% confidence intervals (CI) were retrieved from the final model that demonstrated the best fit according to the Hosmer–Lemeshow goodness-of-fit test. Statistical significance was set at *p* < 0.05 for all analyses.

## 3. Results

Three hundred patients were included in the present cohort, of whom 211 (70.33%) successfully reached the threshold of compliance in at least 70% of evaluated parameters. Differences were noted among institutions that participated in the study in terms of compliance standards, and these seem to be positively related to the actual number of cases that were handled (Table 2, Figure 1). Specifically, large volume centers exhibited the greatest adherence (83.5% and 84.4%, respectively), whereas the three lower-volume centers showed considerably lower adherence (26.9–35.0%). These findings indicate that a substantial portion of the observed inter-hospital variability may reflect differences in caseload and the maturity of ERAS pathway implementation.

Among intraoperative ERAS elements, thromboprophylaxis and compression stockings were uniformly implemented (100%), indicating high compliance and an established adherence. Preadmission education was also provided in the majority of cases (291 patients—97.0%) and did not differ significantly across institutions (*p* = 0.174). On the other hand, avoidance of bowel preparation showed substantial variation (χ^2^ = 172.800, *p* < 0.001), as did administration of antibiotics prior to incision (χ^2^ = 13.198, *p* = 0.010) and preoperative oral carbohydrates (χ^2^ = 65.062, *p* < 0.001). Of note, tranexamic acid use demonstrated one of the highest degrees of institutional variability (χ^2^ = 119.123, *p* < 0.001) (Figure 2).

Intraoperative ERAS elements denoted marked inter-institutional differences. Specifically, preoperative sedative use differed significantly (χ^2^ = 128.880, *p* < 0.001), as did avoidance of epidural/spinal anesthesia (χ^2^ = 81.728, *p* < 0.001) and avoidance of systemic opioids (χ^2^ = 216.890, *p* < 0.001). Use of forced-air warming devices varied across centers (χ^2^ = 11.307, *p* = 0.023). Drain use also differed substantially (χ^2^ = 74.471, *p* < 0.001). Many of these differences were more pronounced in the lower-volume centers, where small sample sizes produced larger proportional shifts (Figure 2).

Similarly, differences were also noted in a large proportion of postoperative ERAS elements, with high-volume centers demonstrating more consistent application of these elements, whereas low-volume centers exhibited greater heterogeneity. Specifically, Foley catheter removal (<24 h) varied by institution (χ^2^ = 27.389, *p* < 0.001), as did the use of postoperative laxatives or chewing gum (χ^2^ = 66.005, *p* < 0.001) and the continuation of intravenous fluids (χ^2^ = 40.690, *p* < 0.001). Likewise, early oral intake on POD 0 (>300 kcal) and POD 1 (>600 kcal) differed markedly across hospitals (χ^2^ = 68.558 and 28.337, respectively; both *p* < 0.001). Finally, early mobilization on POD 0 also showed significant variation (χ^2^ = 43.961, *p* < 0.001) (Figure 2).

Multivariate regression analysis was used to evaluate demographic factors (age, BMI, smoking status), comorbidities (diabetes), ECOG performance status, tumor type, and surgical complexity, which contribute to optimal (>70%) ERAS compliance (Table 3). Of all factors, surgical complexity and ECOG performance status remained independently associated with ERAS adherence after adjusting for BMI, age, smoking, diabetes, and cancer type. Increasing surgical complexity was associated with progressively lower adherence, with intermediate-complexity procedures showing distinctly reduced likelihood of full compliance compared with low-complexity cases, while the highest-complexity category demonstrated an even more extreme effect. Similarly, declining functional status, reflected by ECOG scores, was associated with a stepwise reduction in adherence, with the poorest-performing category showing a profound decrease in the odds of full compliance. Other factors including age, BMI, smoking, diabetes status, and tumor type, did not exhibit statistically meaningful associations. The Hosmer–Lemeshow goodness-of-fit test was suggestive of adequate calibration of the model, an observation that indicates the model’s predictive probability was aligned with the actual distribution of ERAS adherence outcomes. However, in terms of explanatory performance, the Nagelkerke R^2^ test (0.247) indicated that approximately one-quarter of the variance was accounted for by the predictors included in the analysis. Considering that in pragmatic data Nagelkerke R^2^ values rarely exceed 0.3, the actual result (0.247) is evaluated as realistic and appropriate. However, taking into account that it explains a moderate portion of the variability that is observed, one should consider that other factors not accounted in the model also contribute to the observed variance.

Despite the fact that cancer type did not seem to influence the actual decision to optimally implement ERAS parameters, we opted to evaluate the effect in the form of an alluvial plot to better understand how this variable distributes across adherence categories. The alluvial plot demonstrated that the majority of patients with endometrial and ovarian cancer (which constitute the largest diagnostic groups) flow into the “Fulfilled (>70%)” adherence category, as reflected by the broad green and purple streams (Figure 3—right side). On the other hand, patients with less prominent cancer forms (cervical, vulvar, and sarcoma) exhibited a more heterogeneous pattern, with relatively greater proportions of patients flowing into the “Not fulfilled (<70%)” category compared with the larger cancer types.

Considering the results of the multivariate analysis, we constructed an alluvial plot to visualize the patterns of categorical flows between BMI groups, surgical complexity, and ERAS adherence (Figure 3—left side). We observed that normal-weight and overweight patients undergo low- or intermediate-complexity procedures, and these pathways predominantly lead to the “Fulfilled (>70%)” adherence category. Obesity as a factor equally distributes across both surgical complexity categories. It should be noted that a substantial proportion of obese patientsthat had low/intermediate-complexity surgeryultimately receives an optimal ERAS perioperative plan. On the other hand, high-complexity procedures, irrespective of BMI category, seem to flow into the “Not fulfilled (<70%)” category, producing a concentrated purple stream toward this outcome.

## 4. Discussion

### 4.1. Principal Findings

The findings of this prospective multi-institutional cohort of 300 patients undergoing gynecologic oncology surgery within an ERAS framework suggest a high optimal adherence rate (≥70%), as 211 patients (70.33%) reached the predefined threshold for adequate compliance. Implementation of ERAS protocols is expected to vary significantly, particularly when first introduced, with high-volume institutions demonstrating the highest adherence rates; in our cohort, the difference was substantial, as high-volume institutions superseded the necessary threshold easily (83.5% and 84.4%), while the three lower-volume centers exhibited markedly inferior performance, achieving only 26.9% to 35.0% adherence. This pattern suggests that institutional experience, workflow maturity, and cumulative exposure to ERAS practices are key determinants of successful protocol application and indicates that a significant number of cases is needed to help establish a robust protocol with optimal adherence rates. Examination of individual ERAS components revealed a heterogeneous landscape: While some preoperative measures such as thromboprophylaxis, compression stockings, and preadmission education showed near-universal implementation, other elements, including avoidance of bowel preparation, administration of preoperative antibiotics, preoperative carbohydrate loading, and tranexamic acid use, demonstrated significant inter-institutional variability. Intraoperative practices were even more inconsistent, with large disparities noted in the use of preoperative sedation, avoidance of regional anesthesia techniques, minimization of systemic opioids, application of forced-air warming, and drain utilization. Postoperative elements varied similarly across centers, particularly regarding early Foley catheter removal, laxative or chewing gum use, continuation of intravenous fluids, early oral intake on postoperative days 0 and 1, and early mobilization. It should be noted at this point that the components that were more likely to be implanted were those that have gained acceptance in perioperative care, before the implementation of ERAS guidelines, whereas new components that have not been traditionally used were more likely to not gain universal acceptance. The results of our multivariate analysis indicate that the only strong predictors of optimal ERAS adherence were surgical complexity and ECOG performance status. Specifically, increasing operative complexity and poorer functional status were each associated with a progressively lower likelihood of achieving compliance ≥ 70%, an observation that seems reasonable during the first steps of a newly instituted perioperative program, where teams are still adapting workflows, clinical judgment may default on the side of caution in high-risk or technically demanding cases, and the full integration of ERAS principles into complex surgical pathways has not yet reached operational maturity. Neither age, BMI, smoking, diabetes, nor tumor type were significantly associated with adherence. Complementary alluvial visualizations reinforced these findings by illustrating the strong directional flow from high-complexity procedures toward the “Not fulfilled” category, regardless of BMI, and the predominance of endometrial and ovarian cancer patients within the “Fulfilled” adherence group.

### 4.2. Comparison with Existing Literature

Our findings are in accordance with current knowledge as suggested by a recent survey that was conducted across gynecologic oncologists in Europe that denotes widespread implementation of ERAS protocols but substantial variation in adherence to specific elements. They also highlight that, despite formal pathway adoption, key items such as early urinary catheter removal and avoidance of drains remain poorly implemented [23]. Specifically, the overall adherence rate in this survey was comparable to that of our study, ranging between 60 and 80% among the 44% of participating centers. Similar to our study, thromboembolic prophylaxis was highly adopted by the majority of participants (79%), whereas avoidance of bowel preparation demonstrated low compliance rates.

In line with these data, another recent multicenter cross-sectional study from Southwestern China, focusing on gastroenterology, gynecology, hepatobiliary surgery, and urology and published by Mu et al., highlighted that achieving high adherence to ERAS protocols remains challenging, with considerable variability in compliance across institutions [24]. Specifically, the authors of this study demonstrated an overall median adherence of 71.5%, a figure that is also similar to the proportion of patients in our study who achieved ≥70% compliance (70.33%). Nevertheless, in their study, they did not mention a predetermined cut-off value of optimal ERAS adherence, therefore differentiating from our method of reporting. In line with our observation concerning the number of handled cases, Mu et al. also identified treatment in a tertiary center as an independent predictor of higher compliance. These converging observations reinforce the notion that ERAS implementation success is directly related to the level of organizational maturity, the existence of relevant resources, and the actual familiarity and level of training of coordinated multidisciplinary teams. Additionally, both studies reveal significant heterogeneity across individual ERAS elements. Similar to our findings, where certain components such as preadmission education or thromboprophylaxis were nearly universally implemented while others showed marked variation, Mu et al. reported that while adherence to core items such as fasting avoidance exceeded 85%, compliance with components like preoperative carbohydrate loading remained as low as 42%.

Another study published by Nelson et al. that involved five distinct surgical specialties including colorectal, liver, pancreas, gynecologic oncology, and radical cystectomy and evaluated ERAS principles implemented across nine hospital sites demonstrated a gradual increase in adherence rates [25]. The authors reported that the implementation of ERAS pathways reached 76% of cases following proper staff education, a rate that was comparable to that of our study. In this study, the authors observed the feasibility of integrating ERAS beyond single procedures or specialties, aligning with broader efforts to enhance perioperative quality and consistency.

The feasibility of implementing ERAS protocols in highly complex surgical procedures has been implied by Pandraklakis et al. in a cohort of 66 patients undergoing complex cytoreduction for advanced ovarian cancer, with or without the use of hyperthermic intraperitoneal chemotherapy (HIPEC) [19]. Similar results were also reported by a smaller cohort that included 31 patients undergoing cytoreductive surgery and HIPEC that suggested the positive effect of ERAS protocols in this population, together stating that the possibility of increased complications or readmissions was not increased [26]. The most important evidence comes from the PROFAST trial that also targeted high complexity score procedures and reported a surprisingly high overall compliance that reached 92% [27]. It should be noted that in this study, the authors reported no differences in postoperative complications among patients that were randomized in the ERAS arm and control patients. This observation was also confirmed for major complications, including those that referred to the anastomotic site. In our study, a clear negative effect on ERAS protocol compliance was observed with increasing surgical complexity. Keeping in mind the results of the aforementioned studies, it seems that while high compliance is achievable, this can be typically observed among centers with long-standing ERAS culture, robust multidisciplinary coordination, and substantial procedural volume. Although to date, relevant evidence is missing from the literature, multiple barriers may be observed in the institution of ERAS protocols including limited familiarity and confidence among staff, logistical constraints, and reluctance to fully apply early feeding and mobilization, particularly among patients who received multivisceral resections that increase their frailty index.

It should be noted that several of the ERAS elements with the lowest adherence in our cohort were primarily driven by anesthesiology practice. In particular, avoidance of preoperative sedation, minimization of systemic opioids, use of regional anesthesia techniques, maintenance of normothermia, and administration of tranexamic acid all demonstrated marked inter-institutional differences, often among the highest variability observed across the entire pathway, as indicated in Figure 2. Considering that these components are largely controlled by anesthesiologists, their limited engagement likely reflects differing levels of familiarity with ERAS principles, as well as potential variation in training, during the early implementation phase. This interpretation is further supported by the observation that adherence was highest for elements traditionally adopted in routine perioperative workflow, mainly thromboprophylaxis and compression stockings, whereas newly introduced anesthesia-led components showed the lowest uptake. Considering the importance of multidisciplinary engagement to help ensure appropriate ERAS adoption, it seems important to help integrate targeted educational initiatives to establish standardized intraoperative protocols, particularly in centers at the beginning of their ERAS implementation journey.

### 4.3. Strengths and Limitations

The main strength of our study is its prospective, multi-institutional design that captures the real-world implementation of ERAS protocols across gynecologic oncology centers, thereby enhancing the potential external validity and representativeness of reported findings. Its pragmatic design, conducted under routine clinical conditions, helps capture the adoption of ERAS principles in clinical practice across different hospital settings, staffing structures, and resource levels, thereby avoiding the potential constraints of a randomized controlled trial (RCT), which overcomes these barriers through its randomization process. This way, it can provide meaningful insights into real-world barriers to adherence. Another significant strength of our study is its preregistration in ClinicalTrials.gov (NCT06655506), reinforcing its methodological transparency by ensuring adherence to a predefined analytic plan, thus avoiding selective reporting. Moreover, it should be noted that all data were collected using standardized electronic REDCap forms, ensuring procedural consistency, together reducing information bias. The design is anchored in a contemporary, guideline-based ERAS framework following the 2021 ERAS Society recommendations, ensuring alignment with international standards, thus avoiding selection of arbitrary ERAS components. Multivariable logistic regression allowed us to further determine independent predictors of adherence, specifically surgical complexity and ECOG performance status, which were confirmed by the inclusion of alluvial plots that offered an intuitive visualization of interactions among patient factors, surgical complexity, and adherence categories.

On the other hand, it should be noted that the pragmatic nature of the study also means that interventions were not standardized to the degree achievable in an RCT, therefore leaving them prone to the actual variability in institutional workflows, resources, and team experience. Additionally, many participating centers were in the early phases of ERAS adoption, which likely contributed to the significant inter-institutional variability observed, which seems to be more pronounced among anesthesiology-driven components which were less integrated in our series. Moreover, despite the use of two distinct groups that were based on a predefined adherence threshold (≥70% vs. <70%), the observational nature of the study still prevents causal inference. Although the multivariate analysis identified factors potentially associated with optimal ERAS incorporation, the associations identified should be interpreted as descriptive and hypothesis-generating rather than causal. Lastly, clinical outcomes such as length of stay, complications, and readmissions were not evaluated in this interim analysis, as the present sample size could produce underpowered or unstable estimates and risk misleading interpretations. The actual impact of ERAS protocols on these variables will be comprehensively assessed in the completed 600-patient cohort, allowing a more robust determination of how ERAS adherence translates into postoperative recovery quality and overall patient outcomes.

## 5. Conclusions and Implications

The present multicenter study highlights the difficulties of achieving consistent ERAS adherence across gynecologic oncology centers, particularly during the early phases of protocol adoption. The marked variability observed between high- and low-volume centers suggests that implementation success depends on structured protocols, ongoing team training, and the establishment of clear institutional workflows. Our cohort also demonstrated distinct patterns, with adherence being highest in high-volume centers and progressively lower in association with increasing surgical complexity and poorer ECOG performance status. Moreover, anesthesia-driven ERAS elements consistently exhibited the lowest and most heterogeneous uptake across institutions, underscoring discipline-specific barriers during early implementation. The strong influence of surgical complexity and patient performance status on adherence indicates that further familiarization is needed to help physicians constituting the multidisciplinary team confidently apply the ERAS principles in complex scenarios, as current evidence suggests that this is feasible and safe. Future research should focus on identifying practical strategies that can support the adoption of ERAS in settings with limited experience as well as exploring barriers to complex cases. The completion of the full ERGO cohort will allow assessment of temporal improvements in adherence and facilitate evaluation of the relationship between compliance and postoperative outcomes. Additional future stepped-wedge designed cohorts could further clarify causal pathways, something that was not possible in our study, to help guide optimization of instructional procedures and increase the acceptability of ERAS programs in gynecologic oncology.

## Figures and Tables

**Figure 1 cancers-17-03991-f001:**
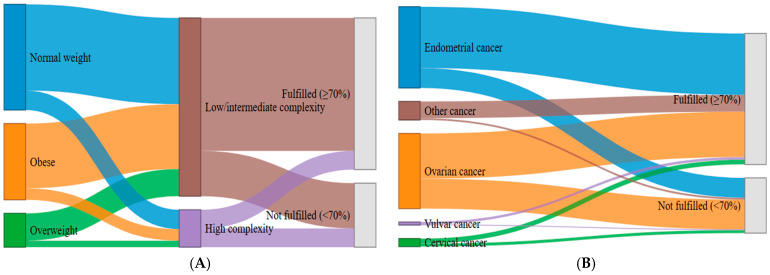
Alluvial flow diagrams illustrating the distribution of patients according to (**A**) BMI category (normal weight, overweight, obesity), surgical complexity (low/intermediate vs. high), and ERAS adherence (fulfilled ≥ 70% vs. not fulfilled < 70%), and (**B**) cancer type (endometrial, ovarian, cervical, vulvar, and other), followed by ERAS adherence. The plots highlight the disproportionate shift of patients undergoing high-complexity procedures toward lower adherence categories and the varying distribution of adherence across cancer types.

**Figure 2 cancers-17-03991-f002:**
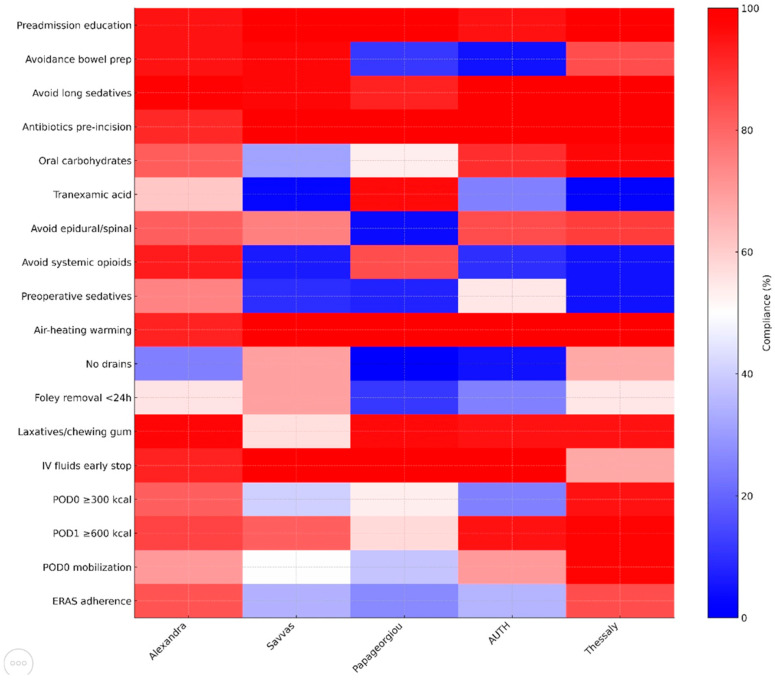
Heatmap depicting adherence (%) to individual ERAS components across participating institutions. A blue–white–red diverging palette reflects low to high compliance. The plot demonstrates substantial inter-institutional variability, particularly in intraoperative and early postoperative items such as avoidance of preoperative sedation, opioid-sparing strategies, oral carbohydrate loading, tranexamic acid use, and early Foley removal, whereas routine preoperative measures show consistently high adherence.

**Figure 3 cancers-17-03991-f003:**
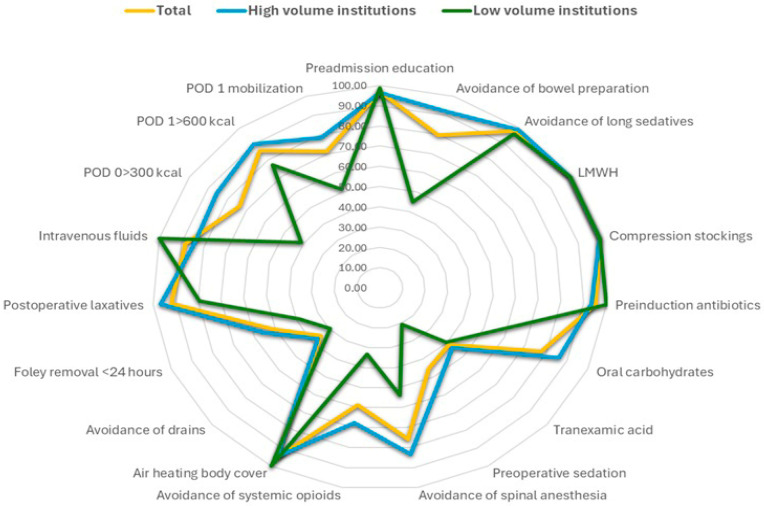
Radar chart comparing adherence (%) to ERAS components across high-volume institutions, low-volume institutions, and the overall cohort. High-volume centers exhibit consistently higher adherence, especially in elements such as avoidance of bowel preparation, sedation minimization, POD1 mobilization, and achievement of postoperative caloric goals. Low-volume centers demonstrate lower and more variable compliance, particularly in intraoperative and early postoperative components, underscoring the influence of surgical volume and institutional experience on ERAS implementation.

**Table 1 cancers-17-03991-t001:** Enhanced recovery protocol components. Components of the Enhanced Recovery After Surgery (ERAS) protocol evaluated in the study, including preoperative, intraoperative, and postoperative elements, with corresponding definitions of compliance.

Phase	ERAS Component	Definition of Compliance
Preoperative	Patient counseling	Written and verbal preoperative information provided, including expectations of early mobilization, oral intake, and discharge criteria.
	Fasting interval	Solids discontinued <6 h and clear fluids allowed up to 2 h preoperatively.
	Carbohydrate loading	Administration of carbohydrate-rich drink 2–3 h before anesthesia induction.
	No mechanical bowel preparation	Absence of preoperative mechanical bowel preparation unless required by procedure (e.g., bowel resection).
	Thromboprophylaxis	Pharmacologic and mechanical prophylaxis according to institutional protocol.
	Antibiotic prophylaxis	Administration within 60 min prior to incision and discontinuation within 24 h postoperatively.
Intraoperative	Multimodal analgesia	Use of regional anesthesia, paracetamol, or NSAIDs in combination with minimized opioid use.
	Maintenance of normothermia	Core body temperature maintained >36 °C intraoperatively.
	Goal-directed fluid therapy	Intraoperative fluids are tailored considering patient’s hemodynamic status.
	No routine use of nasogastric tube	Absence of nasogastric tube beyond recovery period.
	Avoidance of peritoneal drains	No prophylactic intraperitoneal drains used unless clinically indicated.
	Antiemetic prophylaxis	Administration of dual antiemetic regimen (5-HT3 antagonist + dexamethasone or equivalent).
Postoperative	Early oral intake	Initiation of clear fluids within 6 h post-surgery.
	Early mobilization	Patient ambulating or sitting out of bed within 24 h postoperatively.
	Early urinary catheter removal	Catheter removed within 24 h following surgery.
	Multimodal postoperative analgesia	Use of opioid-sparing analgesic regimen, avoiding patient-controlled opioids unless necessary.
	Early resumption of normal diet	Soft or regular diet tolerated within 24 h of surgery.
	Glycemic control	Postoperative blood glucose maintained <180 mg/dL in non-diabetic and diabetic patients alike.
	Fluid balance optimization	Cessation of intravenous fluids within 24 h postoperatively.
	Discharge criteria	Achievement of tolerance to diet, independent ambulation, controlled pain with oral medications, and stable vital signs.

**Table 2 cancers-17-03991-t002:** Compliance rates (and respective %) for each ERAS component across participating institutions. The table highlights significant inter-institutional variability, particularly in intraoperative and early postoperative elements, while routine preoperative items demonstrate higher consistency. (n = patient number).

ERAS Variable	Alexandra Hosp. (n = 158)	St.Savvas Hosp. (n = 32)	Papanikolaou Hosp. (n = 26)	Aristotle Univ. (n = 20)	Univ. of Thessaly (n = 64)	Total (n = 300)	*p*-Value
**Preoperative ERAS components**							
**Preadmission education**	150 (94.9%)	32 (100%)	26 (100%)	19 (95%)	64 (100%)	291 (97.0%)	0.174
**Avoidance bowel prep**	150 (94.9%)	31 (96.9%)	3 (11.5%)	1 (5%)	54 (84.4%)	239 (79.7%)	<0.001
**Avoid long sedatives**	156 (98.7%)	31 (96.9%)	24 (92.3%)	20 (100%)	64 (100%)	295 (98.3%)	0.101
**Antibiotics pre-incision**	144 (91.1%)	32 (100%)	26 (100%)	20 (100%)	64 (100%)	286 (95.3%)	0.010
**Oral carbohydrates**	129 (81.7%)	10 (31.3%)	14 (53.8%)	18 (90%)	62 (96.9%)	233 (77.7%)	<0.001
**Tranexamic acid**	92 (61.3%)	1 (3.1%)	25 (96.2%)	5 (25%)	1 (1.6%)	124 (57.1%)	<0.001
**LMWH prophylaxis**	158 (100%)	32 (100%)	26 (100%)	20 (100%)	64 (100%)	300 (100%)	–
**Compression stockings**	158 (100%)	32 (100%)	26 (100%)	20 (100%)	64 (100%)	300 (100%)	–
**Intraoperative ERAS components**							
**Avoid epidural/spinal**	129 (81.6%)	24 (75%)	1 (3.8%)	17 (85%)	56 (87.5%)	227 (75.7%)	<0.001
**Avoid systemic opioids**	147 (93.0%)	2 (6.3%)	22 (84.6%)	2 (10%)	3 (4.7%)	176 (58.7%)	<0.001
**Preoperative sedatives**	117 (74.5%)	3 (9.4%)	2 (7.7%)	11 (55%)	3 (4.7%)	136 (54.5%)	<0.001
**Air-heating warming**	145 (92.4%)	32 (100%)	26 (100%)	20 (100%)	64 (100%)	287 (96.0%)	0.023
**No drains**	39 (24.7%)	22 (68.8%)	0 (0%)	1 (5%)	43 (67.2%)	105 (35.0%)	<0.001
**Postoperative ERAS components**							
**Foley removal < 24 h**	88 (55.7%)	22 (68.8%)	3 (11.5%)	5 (25%)	35 (54.7%)	153 (51.0%)	<0.001
**Laxatives/chewing gum**	153 (97.5%)	18 (56.3%)	25 (96.2%)	19 (95%)	61 (95.3%)	276 (92.3%)	<0.001
**IV fluids early stop**	144 (92.3%)	32 (100%)	26 (100%)	20 (100%)	43 (67.2%)	265 (88.6%)	<0.001
**POD 0 ≥ 300 kcal**	128 (81.5%)	13 (40.6%)	14 (53.8%)	5 (25%)	61 (95.3%)	221 (73.9%)	<0.001
**POD 1 ≥ 600 kcal**	134 (86.5%)	26 (81.3%)	15 (57.7%)	19 (95%)	63 (98.4%)	257 (86.5%)	<0.001
**POD 0 mobilization**	111 (70.3%)	16 (50%)	10 (38.5%)	14 (70%)	63 (98.4%)	214 (71.3%)	<0.001
**ERAS adherence (%)**	132 (83.5%)	11 (34.4%)	7 (26.9%)	7 (35.0%)	54 (84.4%)	211 (70.3%)	

**Table 3 cancers-17-03991-t003:** Multivariable logistic regression analysis identifying independent predictors of optimal ERAS adherence (>70%). Higher surgical complexity and poorer ECOG performance status were significant negative predictors of adherence. (CI = confidence intervals).

Predictor	Odds Ratio	95% CI	*p*-Value
**Surgical complexity**			
• Intermediate vs. Low	0.263	0.128–0.540	<0.001
• High vs. Low	0.000	0.000	0.999
**BMI status**			
• Overweight vs. normal	0.857	0.377–1.946	0.712
• Obese vs. normal	1.007	0.404–2.510	0.988
• Severly obese vs. normal	1.248	0.542–2.877	0.603
**Age**	1.019	0.992–1.047	0.172
**Smoking status**	0.864	0.441–1.692	0.669
**Diabetes mellitus**	0.934	0.635–1.375	0.731
**Tumor type**			
• Ovarian vs. endometrial	0.786	0.393–1.572	0.496
• Vulvar vs. endometrial	1.327	0.130–13.528	0.811
• Cervical vs. endometrial	0.455	0.125–1.652	0.231
• Sarcoma vs. endometrial	0.263	0.053–1.308	0.103
**ECOG performance status**			
• ECOG 2 vs. 0–1	0.437	0.211–0.906	0.026
• ECOG 3 vs. 0–1	0.032	0.003–0.292	0.002

## Data Availability

The data supporting the findings of this study are not publicly available due to restrictions related to institutional ethical approvals. De-identified individual participant data and the study protocol may be made available upon reasonable request to the corresponding author, subject to approval by the relevant institutional review boards and in accordance with applicable data protection regulations.

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
