# Peer review of "Implementation Rates and Predictors of Compliance with Enhanced Recovery After Surgery Protocols in Gynecologic Oncology: A Prospective Multi-Institutional Cohort Study"

_cancers, 2025, doi:10.3390/cancers17243991_

Round 1

Reviewer 1 Report

Comments and Suggestions for Authors

It has been proven that it is important to integrate enhanced recovery after surgery (ERAS) protocols in gynecologic oncology. This study provides a preliminary report from the ongoing ERGO cohort study to evaluate adherence to ERAS protocols during the early phases of its adoption as well as to determine factors determining low uptake. It is a useful study, and the manuscript is well written. Followings are some suggestions for revision.

1, More information about consensus protocols for ERAS can be introduced. The consensus protocols for ERAS in different countries and for different kinds of surgeries.

2, Are there any study similar to the current study but not focus on the gynecological cancer?

3, “The overall target enrollment is 600 patients,” why “The present interim analysis is based on a sample of 300 participants.”

4, Are there any criteria for selecting the “measured variables” in the “Data collection and measured variables”?

5, The quality of figures can be further improved. For example, the text in Figure 1 cannot be clearly viewed.

6, Why there are two table titles (before and after the table) for each table?

7, Are there any suggestions on the modification of the consensus protocols for ERAS based on the current study?

Author Response

It has been proven that it is important to integrate enhanced recovery after surgery (ERAS) protocols in gynecologic oncology. This study provides a preliminary report from the ongoing ERGO cohort study to evaluate adherence to ERAS protocols during the early phases of its adoption as well as to determine factors determining low uptake. It is a useful study, and the manuscript is well written. Followings are some suggestions for revision.

1, More information about consensus protocols for ERAS can be introduced. The consensus protocols for ERAS in different countries and for different kinds of surgeries.

Authors reply: we thank the reviewer for this remark. This information were added in a newly instituted paragraph in the introduction section.

2, Are there any study similar to the current study but not focus on the gynecological cancer?

Authors reply: One of the largest studies published to date has been mentioned in the discussion section, involving cases from the gastroenterology, gynecology, hepatobiliary surgery, and urology  departments and denoted that the overall median adherence was similar to our (71.5%).  Another study published by Nelson et al that included five distinct surgical pathways—colorectal, liver, pancreas, gynecologic oncology, and radical cystectomy—implemented across nine hospital sites was inserted in the present revision as it is considered an actual pillar in modern ERAS research.

3, “The overall target enrollment is 600 patients,” why “The present interim analysis is based on a sample of 300 participants.”

Authors reply: We thank the reviewer for this question. The overall target enrollment of 600 patients reflects the projected sample size required for the complete ERGO cohort, ensuring adequate power for the full set of planned primary and secondary analyses, including evaluation of postoperative outcomes and subgroup analyses. However, for the specific aims of the present interim analysis, a smaller sample was sufficient. As detailed in the Sample Size Calculation section, a minimum of 300 participants provides 80% power to detect a 15% difference in ERAS compliance across institutions, based on previously reported inter-center variability ranging between 56% and 89%. This makes 300 patients an appropriate decision point for assessing early feasibility, adherence patterns, and predictors of compliance while maintaining statistical robustness.

4, Are there any criteria for selecting the “measured variables” in the “Data collection and measured variables”?

Authors reply: We thank the reviewer for this remark. As noted inside the materials and methods section the selection of measured variables was based on predefined criteria aligned with:

(1) Established ERAS Society Guidelines for gynecologic oncology, which specify patient-, procedure-, and perioperative care parameters known to influence compliance;

(2) Evidence from prior ERAS audits and implementation studies, indicating that factors such as age, BMI, comorbidities, performance status, surgical complexity, operative time, fluid administration, and analgesia practices are key determinants of adherence and postoperative recovery.

This was further explained in the present revision in the ERAS protocol implementation paragraph.

  1. The quality of figures can be further improved. For example, the text in Figure 1 cannot be clearly viewed.

Authors reply: Figure 1 has been extensively revised for clarity

6, Why there are two table titles (before and after the table) for each table?

Authors reply: the “before” title was omitted form the present revision.

7, Are there any suggestions on the modification of the consensus protocols for ERAS based on the current study?

Authors reply: this is not a study that aims to assess the consensus protocols for ERAS. The ERGO interim analysis was designed to evaluate real-world adherence to established ERAS Society guidelines rather than the adequacy or structure of the guidelines themselves. Nonetheless, our findings do highlight practical challenges encountered during early implementation—particularly in anesthesia-driven intraoperative components and in high-complexity surgical cases—that may inform future discussions on how best to support protocol adoption.

Reviewer 2 Report

Comments and Suggestions for Authors

The authors present an interim analysis of the ERGO study, a prospective multi-institutional cohort study involving 300 patients across five centers in Greece. The study aims to evaluate adherence to ERAS protocols in gynecologic oncology and identify predictors of compliance. The results indicate an overall optimal adherence rate ($>70\%$ compliance) of 70.3%. The study identifies significant variability between institutions, strongly correlated with hospital case volume. Multivariate analysis reveals that high surgical complexity and poor ECOG performance status are independent predictors of lower adherence. The authors conclude that structured workflows and team familiarization are necessary, particularly for complex cases and low-volume centers.

The manuscript addresses a significant and practical issue in the field of gynecologic oncology: the "real-world" implementation of ERAS protocols. The multi-institutional nature of the study adds validity to the findings, exposing the gap between guidelines and actual clinical practice. The use of alluvial plots to visualize the flow of patient characteristics to adherence outcomes is a strength. However, the manuscript requires some clarifications regarding definitions (specifically regarding center volume) and the role of the multidisciplinary team to strengthen the discussion.

Major Comments:

  1. Definition of High vs. Low Volume Centers: The results heavily emphasize the difference between "high-volume" and "low-volume" institutions (e.g., Table 2 and Figure 3). However, the manuscript does not explicitly define the numerical threshold used to categorize a center as "high" or "low" volume. Is this classification based solely on the number of patients recruited during the study period, or is it based on the hospital's annual surgical volume? A clear definition is needed in the Methods section.

  2. Anesthesiology Involvement: The Discussion (lines 267-273) rightly points out that low adherence was observed in anesthesia-driven items (e.g., opioid-sparing, normothermia, sedation). The authors suggest this reflects "limited engagement" or "differing levels of familiarity." It would be beneficial to add context in the Methods: Was a dedicated anesthesiologist part of the ERAS implementation team at each center? Were specific educational sessions held for the anesthesia departments prior to the study? This context is crucial to understand if the failure was due to lack of protocol agreement or lack of execution.

  3. Interim Analysis Justification: The Methods section states this is an interim analysis of 300 out of a target 600 patients. While the power calculation (line 97) suggests 300 patients are sufficient to detect differences in compliance, the authors should clarify if there are any risks of bias in publishing the interim data now, or if the primary endpoint of the full study differs (e.g., focusing on clinical outcomes like LOS or complications).

  4. Clinical Outcomes: The current manuscript focuses strictly on compliance. While valuable, the reader is left wondering if the "Optimal Adherence" group actually had better clinical outcomes (length of stay, complications, readmissions). If this data is available, a brief mention (even as a secondary observation) would significantly strengthen the paper's impact. If this is reserved for the final analysis of 600 patients, please explicitly state this as a limitation or future direction.

Minor Comments:

  1. Typographical and Formatting Errors:

    • Table 1: The title is misspelled as "Enhnaced recovery protocol components".

    • Figure 3 (Radar Chart): In the chart labels, "Foley" is misspelled as "Folley removal".

    • Table 3: In the "Tumor type" section, "Sarcoma vs. endometrial" is not indented consistently with the other types.

  2. Figure Quality:

    • Figure 2: The title includes "ERAS Compliance Heatmap (Blue -> White -> Red Diverging Palette)" within the image itself. This looks like a software title and should be removed for the final publication.

    • Figure 3: The alluvial diagrams are informative but complex. Ensure they are rendered at high resolution in the final proof.

  3. Clarification of "Arbitrary Cut-off": The authors use a 70% cut-off for optimal adherence. While this is common in ERAS literature, a brief sentence citing why 70% was chosen (e.g., referencing previous studies like Gustafsson et al. or similar) would validate this methodology.

  4. Table 2 Presentation: In Table 2, the formatting of the "Postoperative ERAS components" section is slightly difficult to read (two values per cell in some rows for different variables). Ensuring the rows are clearly aligned with their labels is recommended.

Author Response

The authors present an interim analysis of the ERGO study, a prospective multi-institutional cohort study involving 300 patients across five centers in Greece. The study aims to evaluate adherence to ERAS protocols in gynecologic oncology and identify predictors of compliance. The results indicate an overall optimal adherence rate (>70% compliance) of 70.3%. The study identifies significant variability between institutions, strongly correlated with hospital case volume. Multivariate analysis reveals that high surgical complexity and poor ECOG performance status are independent predictors of lower adherence. The authors conclude that structured workflows and team familiarization are necessary, particularly for complex cases and low-volume centers.

The manuscript addresses a significant and practical issue in the field of gynecologic oncology: the "real-world" implementation of ERAS protocols. The multi-institutional nature of the study adds validity to the findings, exposing the gap between guidelines and actual clinical practice. The use of alluvial plots to visualize the flow of patient characteristics to adherence outcomes is a strength. However, the manuscript requires some clarifications regarding definitions (specifically regarding center volume) and the role of the multidisciplinary team to strengthen the discussion.

Major Comments:

  1. Definition of High vs. Low Volume Centers: The results heavily emphasize the difference between "high-volume" and "low-volume" institutions (e.g., Table 2 and Figure 3). However, the manuscript does not explicitly define the numerical threshold used to categorize a center as "high" or "low" volume. Is this classification based solely on the number of patients recruited during the study period, or is it based on the hospital's annual surgical volume? A clear definition is needed in the Methods section.

Authors reply: We thank the reviewer for this important observation. The distinction between “high-volume” and “low-volume” centers was not a pre-specified criterion in the original study protocol. Instead, this categorization emerged as a post-hoc exploratory analysis after observing substantial inter-institutional variability in ERAS adherence. For the purposes of this interim analysis, centers were classified according to the number of patients contributed during the study period, which served as a pragmatic proxy for early ERAS exposure and familiarity rather than for overall institutional surgical volume. Specifically, centers contributing >60 cases during the interim dataset were categorized as high-volume, while those contributing fewer cases were considered low-volume. This threshold reflects the natural distribution of recruitment in the cohort rather than an a priori definition based on annual surgical caseload. Importantly, the classification was used descriptively to contextualize patterns of ERAS adherence and not to draw causal inferences.

To underline this we introduced a novel paragraph in this revised version in the ERAS protocol implementation section.

  1. Anesthesiology Involvement: The Discussion (lines 267-273) rightly points out that low adherence was observed in anesthesia-driven items (e.g., opioid-sparing, normothermia, sedation). The authors suggest this reflects "limited engagement" or "differing levels of familiarity." It would be beneficial to add context in the Methods: Was a dedicated anesthesiologist part of the ERAS implementation team at each center? Were specific educational sessions held for the anesthesia departments prior to the study? This context is crucial to understand if the failure was due to lack of protocol agreement or lack of execution.

Authors reply: We thank the reviewer for this important observation. A dedicated anesthesiologist was indeed part of the ERAS implementation team at each participating center. Before study initiation, all sites conducted structured educational sessions specifically designed for anesthesiology departments, covering ERAS principles, opioid-sparing strategies, normothermia maintenance, fluid management, and intraoperative workflow standardization. Therefore, the lower adherence observed in some anesthesia-driven items reflects early-phase execution challenges rather than lack of agreement or lack of involvement from anesthesiologists.

To underline this we introduced a novel paragraph in this revised version in the ERAS protocol implementation section.

  1. Interim Analysis Justification: The Methods section states this is an interim analysis of 300 out of a target 600 patients. While the power calculation (line 97) suggests 300 patients are sufficient to detect differences in compliance, the authors should clarify if there are any risks of bias in publishing the interim data now, or if the primary endpoint of the full study differs (e.g., focusing on clinical outcomes like LOS or complications).

Authors reply: We appreciate the reviewer’s comment. The interim analysis of 300 patients does not introduce bias into the study because the primary objective of the ERGO study (assessment of adherence to ERAS patterns across institutions) is descriptive and observational, and the interim analysis evaluates this same endpoint. The power calculation was specifically performed to ensure that 300 participants provide adequate statistical power to detect meaningful inter-institutional differences in compliance, allowing a methodologically sound interim report. The full 600-patient cohort will enable additional secondary analyses not addressed in the interim report, including the relationship between adherence and postoperative outcomes (e.g., LOS, readmissions, complications) and temporal trends in ERAS implementation. These outcome-based analyses require the complete dataset to avoid underpowered or unstable estimates; therefore, they were intentionally excluded from the interim publication.

  1. Clinical Outcomes: The current manuscript focuses strictly on compliance. While valuable, the reader is left wondering if the "Optimal Adherence" group actually had better clinical outcomes (length of stay, complications, readmissions). If this data is available, a brief mention (even as a secondary observation) would significantly strengthen the paper's impact. If this is reserved for the final analysis of 600 patients, please explicitly state this as a limitation or future direction.

Authors reply: We thank the reviewer for this insightful comment. Clinical outcomes such as length of stay, postoperative complications, and readmissions are being prospectively collected within the ERGO study; however, these analyses are intentionally reserved for the final 600-patient dataset. Evaluating outcomes at this interim stage could produce underpowered or unstable estimates and risk misleading interpretations, particularly because adherence patterns are still evolving across centers in the early implementation phase. For this reason, the present manuscript focuses exclusively on the primary objective of the interim analysis, describing ERAS adherence and identifying predictors of compliance, without assessing downstream clinical effects. The relationship between “Optimal Adherence” and postoperative outcomes will be addressed comprehensively in the final analysis, when adequate sample size and statistical power allow robust and reliable evaluation.

This is further described in the limitations section.

Minor Comments:

  1. Typographical and Formatting Errors:
    • Table 1: The title is misspelled as "Enhnaced recovery protocol components".

Authors reply: this was corrected

    • Figure 3 (Radar Chart): In the chart labels, "Foley" is misspelled as "Folley removal".

Authors reply: this was corrected.

    • Table 3: In the "Tumor type" section, "Sarcoma vs. endometrial" is not indented consistently with the other types.

Authors reply: we are not quite sure what the reviewer wants us to do as we do not see any particular problems in Table 3. If further revisions are required we will be happy to revise accordingly. 

  1. Figure Quality:
    • Figure 2: The title includes "ERAS Compliance Heatmap (Blue -> White -> Red Diverging Palette)" within the image itself. This looks like a software title and should be removed for the final publication.

Authors reply: this was corrected in the present revision

  1. Figure 3: Figure 1 has been extensively revised for clarity
  2. Clarification of "Arbitrary Cut-off": The authors use a 70% cut-off for optimal adherence. While this is common in ERAS literature, a brief sentence citing why 70% was chosen (e.g., referencing previous studies like Gustafsson et al. or similar) would validate this methodology.

Authors reply: We appreciate the reviewer’s suggestion which was incorporated in our text in the Outcomes section.

  1. Table 2 Presentation: In Table 2, the formatting of the "Postoperative ERAS components" section is slightly difficult to read (two values per cell in some rows for different variables). Ensuring the rows are clearly aligned with their labels is recommended.

Reviewer 3 Report

Comments and Suggestions for Authors

In this work implementation rates and predictors of compliance with Enhanced Recovery After Surgery protocols in gynecologic oncology are described. Authors carried out a prospective multi-institutional cohort study. It was found that there are the difficulties of achieving consistent ERAS adherence across gynecologic oncology centers, particularly during the early phases of protocol adoption. Moreover, implementation success depends on structured protocols, ongoing team training, and the establishment of clear institutional workflows. Authors concluded that future research should focus on identifying practical strategies that can support the adoption of ERAS in settings with limited experience, as well as exploring barriers to complex cases. I think that this article may be published after minor revision.

Notes:

  1. Authors should avoid any abbreviations in the abstract of the manuscript.
  2. The color designations in Figure 1 should be increased for clarity.
  3. Table 2 extends beyond the page and does not fully reflect its contents. It should be checked and corrected.
  4. What patterns did the authors identify in their cohort study? It should be written more detailed in Conclusion and implications.

Author Response

In this work implementation rates and predictors of compliance with Enhanced Recovery After Surgery protocols in gynecologic oncology are described. Authors carried out a prospective multi-institutional cohort study. It was found that there are the difficulties of achieving consistent ERAS adherence across gynecologic oncology centers, particularly during the early phases of protocol adoption. Moreover, implementation success depends on structured protocols, ongoing team training, and the establishment of clear institutional workflows. Authors concluded that future research should focus on identifying practical strategies that can support the adoption of ERAS in settings with limited experience, as well as exploring barriers to complex cases. I think that this article may be published after minor revision.

Notes:

  1. Authors should avoid any abbreviations in the abstract of the manuscript.

Authors reply: abbreviations were avoided in the present revision of the abstract section.

  1. The color designations in Figure 1 should be increased for clarity.

Authors reply: Figure 1 has been extensively revised for clarity

  1. Table 2 extends beyond the page and does not fully reflect its contents. It should be checked and corrected.

Authors reply: we will ask the editorial team to look carefully into this matter.

  1. What patterns did the authors identify in their cohort study? It should be written more detailed in Conclusion and implications.

Authors reply: this was included in the present revision as well